# Addressing vaccine hesitancy in developing countries: Survey and experimental evidence

**Christopher Hoy** [1]*, **Terence Wood**[2], **Ellen Moscoe**[3]

**1** Poverty and Equity Group, The World Bank, Washington, DC, United States of America, **2** Development Policy Centre, Crawford School of Public Policy, The Australian National University, Canberra, Australia, **3** Mind Behavior and Development Unit, The World Bank, Washington, DC, United States of America

* choy@worldbank.org

## Abstract

Vaccine hesitancy is proving to be a significant impediment to COVID-19 vaccination campaigns in some developing countries. This study focuses on vaccine hesitancy and means of reducing it. Data come from a large, representative phone survey and online randomized survey experiment, both run in Papua New Guinea, a developing country with low vaccination rates. Less than 20% of relevant respondents to the phone survey were willing to be vaccinated, primarily because of fear of side effects and low trust in the vaccine. Although vaccine hesitancy was high in the online experiment, participants who received a message emphasizing that the vaccine was safe and COVID-19 dangerous were 68% more likely to state they planned to be vaccinated than those in the control group. A message appealing to social norms was also effective in reducing vaccine hesitancy, although its efficacy was limited to certain types of people.

## Introduction

High vaccination rates are an essential component in managing the COVID-19 pandemic. Initially, the limited supply of vaccines, especially in low- and middle-income countries, was a major constraint on vaccination programs [1]. As vaccines started to become more readily available throughout 2021, it became clear that limited demand also posed a significant challenge in some developing countries, with people unwilling to be vaccinated against COVID-19 even when provided the opportunity [2–4]. This raises the question of what governments and aid donors can do to reduce people's hesitancy about receiving a COVID-19 vaccine in these settings.

In this paper we report on the findings of research studying drivers of vaccine hesitancy as well as research on ways of increasing people's willingness to receive a COVID-19 vaccine. Our data come from a combination of a large representative phone survey and an online randomized survey experiment in Papua New Guinea (PNG). With a population of nearly 9,000,000 people in 2020, PNG is more populous than the median state in the Asia Pacific Region [5]. Despite significant problems with the virus, PNG's COVID-19 vaccination rates are among the lowest in the world [6]. Reports of protest and even violence have dramatically illustrated the problem of vaccine hesitancy in parts of the country [7–9].

**Data Availability Statement:** If the manuscript is accepted for publication, we will promptly upload data files on the Harvard Dataverse (https://dataverse.harvard.edu/), and provide you with the DOIs for the files.

**Funding:** Funding for this study was provided by Gavi, the Vaccine Alliance; the Korea Trust Fund for Economic and Peace-Building Transitions; and The United States Agency for International Development (through the Papua New Guinea UNICEF office). The funders played no role in the the study design, data collection and analysis, decision to publish, or preparation of the manuscript.

**Competing interests:** The authors have declared that no competing interests exist.

Existing published research on attitudes to COVID-19 vaccine hesitancy in developing countries has been limited to observational surveys which have sought to quantify its prevalence. These surveys have covered 15 African countries [2], as well as a group of developing countries globally [3]. The studies have found hesitancy varies significantly across developing countries, although it is prevalent enough to impede fully successful vaccination programs in many countries. Existing published experimental research on reducing COVID-19 vaccine hesitancy comes solely from developed countries and has returned mixed results. Some experiments have successfully changed views, while others have not, or have only found certain treatments to be effective [10–15].

Although the specific study of COVID-19 vaccine hesitancy is still comparatively new, a rich body of work has accumulated focused on people's unwillingness to be vaccinated against other diseases. This research has taken the form of both observational studies and experimental work. A common finding from many observational studies of attitudes to vaccinations is that people's reluctance to be vaccinated stems in part from fear of vaccines or their perceived side effects [16–18]. Similarly, at least some observational studies have found that people who perceive an illness to be threatening, and vaccines efficacious in reducing that threat, are more willing to be vaccinated [18, 19].

Although observational studies provide good grounds for anticipating that information on the relative risk of vaccines and diseases will increase people's willingness to be vaccinated, experimental studies have returned mixed results. Positive treatment effects have been found in some studies [19], other experiments have failed to find clear evidence of the efficacy of information treatments [17, 20], while some studies have found adverse effects, at least in certain groups [21, 22].

The range of findings from this broader literature emphasizes the importance of learning more about attitudes to vaccines for COVID-19. It cannot simply be assumed that awareness campaigns providing the public with information will change attitudes to vaccinations. Our work provides one of the first detailed descriptions of COVID-19 vaccine hesitancy using both descriptive and experimental evidence from a developing country. In doing this we make a notable contribution to the existing literature on vaccine hesitancy. Focusing on a developing country where vaccination rates are very low, we identify the most plausible drivers of hesitancy regarding the COVID-19 vaccine in this context, and we demonstrate that COVID-19 vaccine hesitancy can potentially be reduced.

## Material and methods

Much of PNG is remote and parts of the country are prone to violence. Given these challenges, and the need to learn about vaccine hesitancy in a timely manner, with minimal in person contact, phone and online surveys were the most appropriate means of studying attitudes to COVID-19.

### Phone survey

To collect information about levels of vaccine hesitancy and what factors might be contributing to it, a phone survey covering 2,533 households was conducted from May 26 to June 6, 2021. To address the potential concern that phone survey data would be un-representative we undertook three steps. First, we used quotas based on phone tower locations to ensure people we sampled were from across the country. Second, we oversampled phone users whose phone use patterns in cell phone company data matched those of poorer users as identified from other recent studies in PNG. Third, we generated survey weights by combining survey

participants' responses to sociodemographic questions with data from the most recent census (see S1 File).

The phone survey asked whether people were planning to be vaccinated as well as questions about potential drivers of COVID-19 vaccine hesitancy (see S2 File).

The data collected from the phone survey were both summarized descriptively and analyzed using multiple regression analysis. The regression analysis involved using OLS regressions to estimate the effect on willingness to be vaccinated (the dependent variable) from a dummy variable for a given potential driver of vaccine hesitancy (the independent variable) after controlling for demographic characteristics of respondents.

## Survey experiment

The online randomized survey experiment was conducted via Facebook from June 24 to July 26, 2021. Participants were recruited using Facebook advertisements and taken to a chatbot in Facebook messenger to complete the survey. People were only recruited if they were over 18 (as per Facebook's data; they were also explicitly asked their age prior to the survey commencing). The sample was drawn from all Facebook accounts in PNG and quotas were applied based on demographic characteristics (age, gender and regions) to ensure adequate representation in the sample. It is estimated that almost 20% of the adult population in PNG have Facebook accounts [23]. To make the survey data as representative as possible of the overall population, the data were weighted using inverse probability weights based on age, sex and location calculated using the most recent census (see S1 File).

The treatments in the online experiment were based on existing work on vaccine hesitancy, but were also tailored to suit the PNG context based on consultation with research partners in PNG. The resulting three treatments (expert advice, social norms and relative safety) are shown in Table 1.

Almost 2,000 people participated in the survey experiment and respondents were randomly allocated into four groups (the three treatment groups and the control group). This allowed for cross-sectional analysis between the treatment and control groups. Forty percent of participants were allocated to the control group and the remaining 60% being split between the three treatment groups. Questions about vaccination intentions were not asked of people who were already vaccinated. With the available sample it was possible to detect a treatment effect in the order of 8 to 10 percentage points. (This is based on a statistical power calculation with an alpha of 0.05 and beta of 0.2.)

Randomization ensured there were very few statistically significant differences between treatment and control groups across demographic characteristics (S4 Table). As a result, we report on treatment effects from simple across group comparisons in the body of this paper. In

**Table 1. Design of randomized survey experiment.**

| Group | Question | n |
|---|---|---|
| Control | Do you plan to get the COVID-19 vaccine? | 741 |
| Treatment A—expert advice | The COVID vaccines available in PNG are considered safe and highly effective by national and international experts. Do you plan to get the COVID-19 vaccine? | 416 |
| Treatment B—social norms | Papua New Guineans are getting vaccinated against COVID-19! More than 20,000 have done it so far. Help us protect our communities! Do you plan to get the COVID-19 vaccine? | 361 |
| Treatment C—relative safety | COVID-19 vaccines are safe—there have been no severe side effects reported, compared to hundreds of deaths due to COVID-19 in PNG. Do you plan to get the COVID-19 vaccine? | 382 |

S3 Table in the supporting information we report on regression analysis with controls added. Results are effectively the same.

Both the phone survey and the online experiment were conducted in Tok Pisin, PNG's lingua franca. The survey was approved by the World Bank's Ethical Conduct Review Board and data collected in line with World Bank protocols regarding research involving human participants. Participants in both surveys only participated after being fully informed of what engagement meant using standard text provided by the World Bank Ethical Conduct Board. Participants indicated their willingness to take part in the phone survey verbally. Consent for the experiment involved explicitly opting into participating in the experiment online.

Once data were provided to us from the surveys they were managed, tidied and analyzed in Stata IC 14.1. The threshold for statistically significant findings reported in the paper is p<0.05.

## Results

### Vaccine hesitancy

The phone survey revealed very high levels of vaccine hesitancy in PNG. Only a small minority (18%) of Papua New Guineans who had heard a vaccine was available were planning on being vaccinated. Most respondents stated they were unwilling to be or not sure about being vaccinated against COVID-19 (see Fig 1).

In the phone survey, the main reason people provided when asked why they did not want to be (or were unsure about being) vaccinated was that they were worried about the side effects of the COVID-19 vaccine (see Fig 2). The second most common response was low trust in the vaccine.

The phone survey also asked a series of questions about factors that previous work has suggested may be related to vaccine hesitancy [2, 17, 20, 24]. (See S2 File for a list of survey questions.) Factors included respondents' beliefs about whether they trusted the vaccine, the

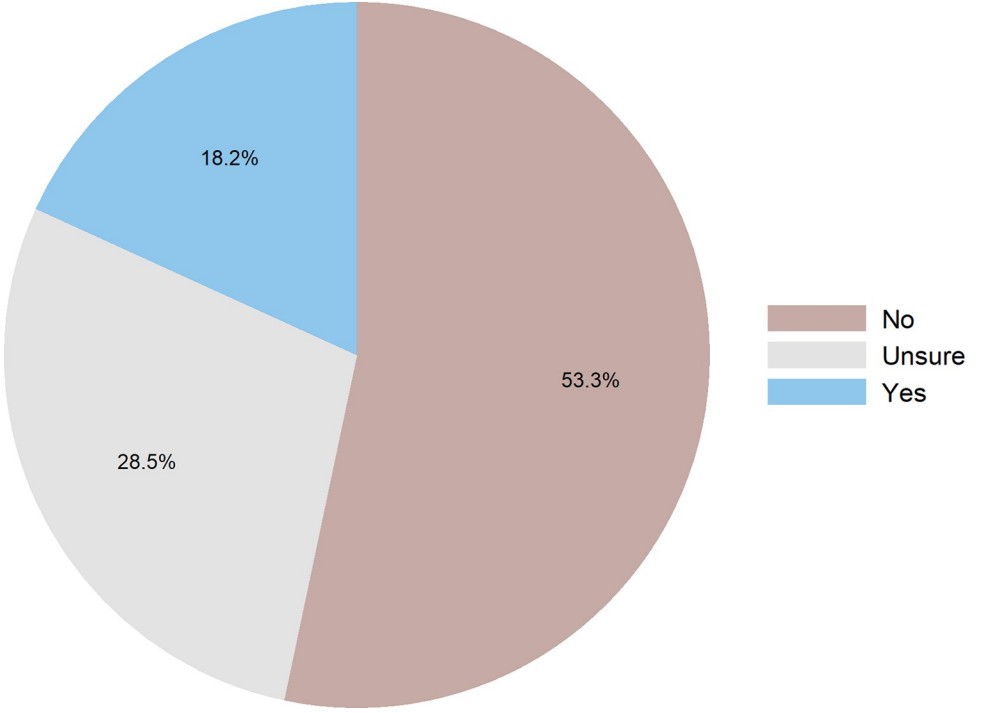

**Fig 1. Levels of vaccine hesitancy in PNG from phone survey.**

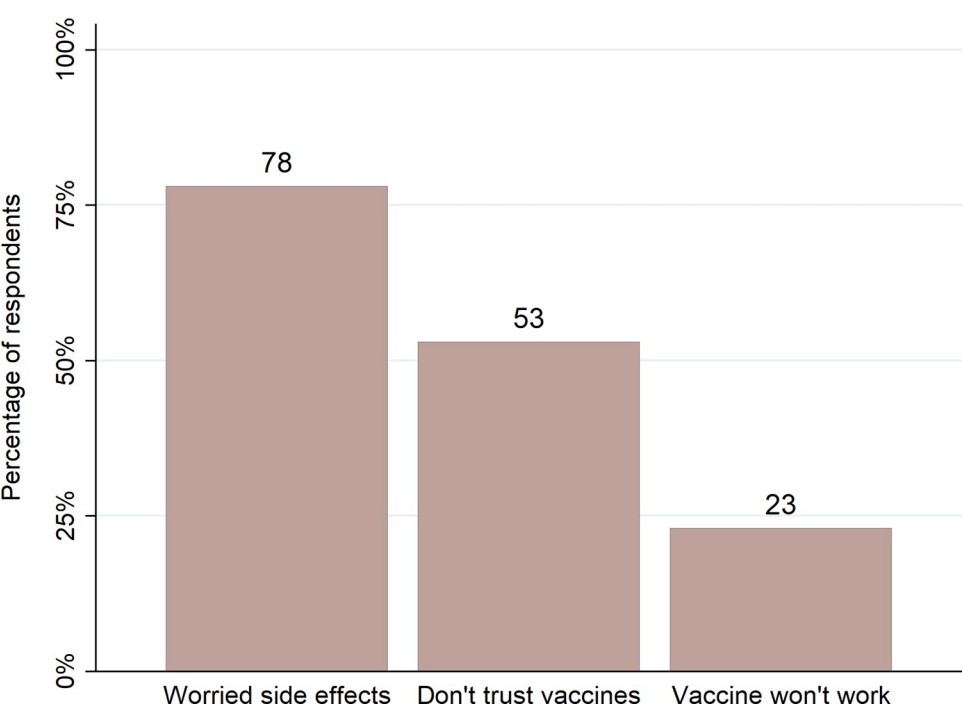

**Fig 2. Top three reasons for vaccine hesitancy in PNG.** Note: people could select more than one reason. As a result, responses total to more than 100%.

behavior of their family and friends, their concern about getting COVID-19 and their prior history of vaccination. Descriptive analysis of the survey data shows that respondents who trusted the vaccine and who believed that their family and friends would be vaccinated were much more likely to be willing to get the vaccine, whereas no such pattern exists for other potential factors. This finding is summarized in Fig 3, which is based on separate ordinary least squares (OLS) regressions comparing willingness to be vaccinated and each factor individually while controlling for respondents' demographic characteristics. Very similar patterns exist when variables for all potential factors are included in the same regression (along with the demographic characteristics) and when examining the bivariate correlations between vaccine hesitancy and these factors individually (S1 and S2 Tables).

## Reducing vaccine hesitancy

Although vaccine hesitancy was high, phone survey results suggested people might plausibly be amenable to having their views changed (see Fig 4). Respondents that did not plan on being vaccinated or were unsure were asked, "Would you be more likely to receive the COVID-19 vaccine if any of the following individuals/authorities receive or recommend the vaccine?" Only 34% of respondents stated that no one could change their mind, whereas 48% of respondents said that recommendations from health workers would make them more likely to be vaccinated. In addition, phone survey respondents were asked their most preferred way of receiving information about the vaccine and the vast majority of respondents (77%) stated face-to-face communication from health workers (Fig 4).

## Survey experiment

Participants in the online randomized survey experiment were randomly allocated to one of four groups: the control group, which received no message, but which was simply asked about

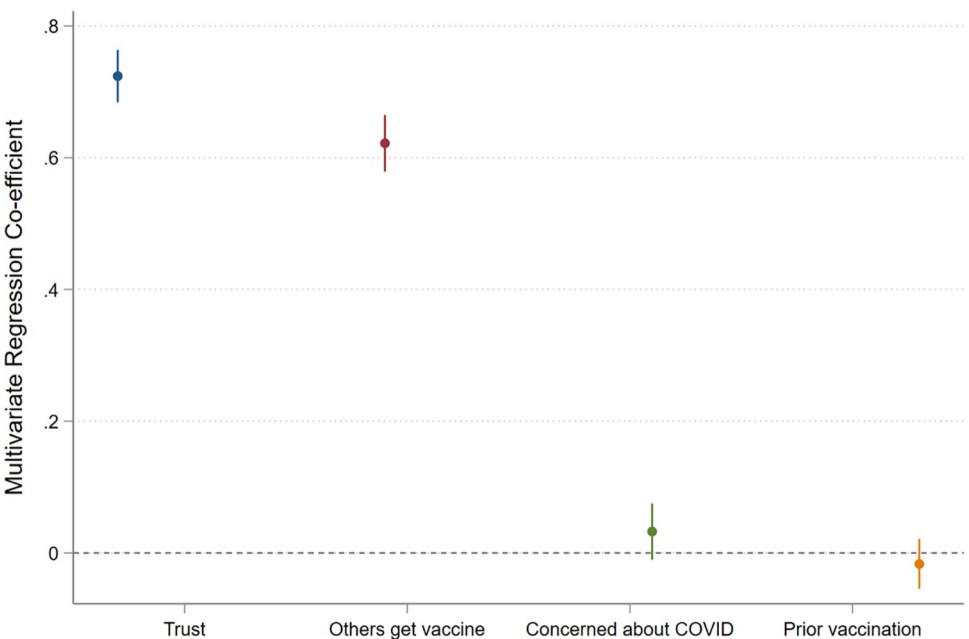

**Fig 3. Potential drivers of vaccine hesitancy and willingness to be vaccinated against COVID-19.** Note: Coefficients are shown from regression models in which demographic factors are controlled for. 95 percent confidence intervals are shown with error bars.

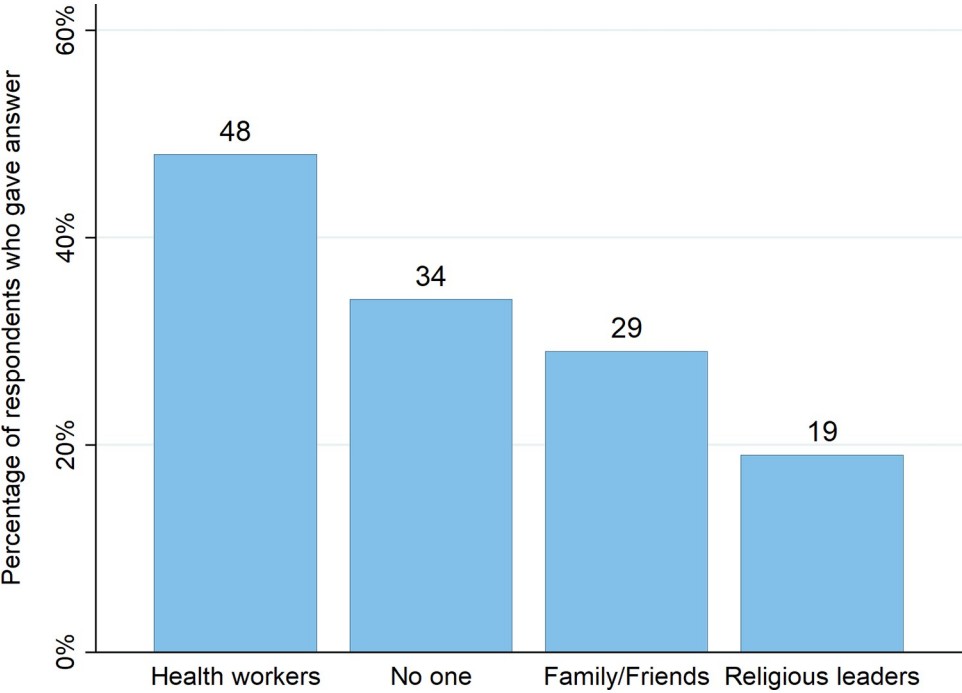

**Fig 4. Top three groups in people who could change respondents' minds about vaccination.** Note: people could provide more than one response. As a result, responses total to more than 100%.

vaccine intentions; the 'expert advice' treatment group that was informed that international and national experts endorsed the vaccine; the 'social norms' group, which was told that tens of thousands of people in PNG were being vaccinated; and the 'relative safety' group which was told the vaccine was safe and that COVID-19 was dangerous.

Fig 5 shows the central results of the survey experiment. Among the treatments, the relative safety treatment clearly increased people's willingness to be vaccinated. Only 17.1% of respondents in the control group were willing to be vaccinated, whereas 28.7% of respondents in the relative safety treatment group were willing to be vaccinated. The social norms treatment also increased the share of respondents willing to be vaccinated by 9.4 percentage points. However, the expert advice treatment did not have a statistically significant effect. After controlling for demographic characteristics of respondents, the impact of the relative safety and the social norms treatments were still large and statistically significant (S3 Table).

The relative safety treatment had a similar effect in reducing the share of respondents' who were certain they would not be vaccinated. Almost 30% (28.2%) of respondents in the control group stated they were unwilling to be vaccinated whereas this figure was only 18.8% in the relative safety treatment group. The social norms treatment also reduced people's unwillingness to be vaccinated by 7.3 percentage points. There was no significant effect from the expert advice treatment. Similar results were found after controlling for the demographic characteristics of respondents (S3 Table).

Both with respect to increasing the share of people who said they would be vaccinated, and reducing the share who said they would not be vaccinated, the relative safety treatment had statistically significant effects across most of the demographic groups of respondents (S2 Fig). The social norms treatment was only clearly effective on women, rather than men. As a consequence, and the fact that women were under-represented in the survey experiment, the social norms treatment was only statistically significant when in the survey data was weighted to match the general population. The expert advice treatment was not statistically significant in any demographic group.

## Discussion

Vaccine hesitancy is proving to be a serious impediment to COVID-19 vaccination programs in some developing countries. Not only does this increase the health burdens these countries face, but–to the extent it increases global numbers of unvaccinated people–hesitancy in developing countries adds to the risk of new variants of the virus emerging and to the ongoing global challenge of the pandemic [4].

Our study highlights three key areas for policymakers to prioritize as they attempt to reduce vaccine hesitancy. First, it provides evidence that boosting people's trust in COVID-19 vaccines and reducing their fear of adverse effects from vaccination appears to be integral in increasing people's willingness to be vaccinated. The phone survey showed both low trust in the vaccine and high fear of side effects. There was also a strong correlation amongst respondents between willingness to be vaccinated and trust in the vaccine. The online experiment illustrated that information about the relative safety of COVID-19 vaccines substantially increases willingness to be vaccinated and reduces opposition to receiving the vaccine.

Second, the importance of social norms in shaping people's willingness to be vaccinated was illustrated by the effects of the social norms treatment and reaffirmed by phone survey findings showing a clear relationship between vaccine intentions and people's beliefs about whether their family and friends would be vaccinated. At present it appears that PNG is in a low-level equilibrium whereby few people are willing to be vaccinated, in part because few people have been vaccinated. Although PNG is clearly a long way from transitioning into a high

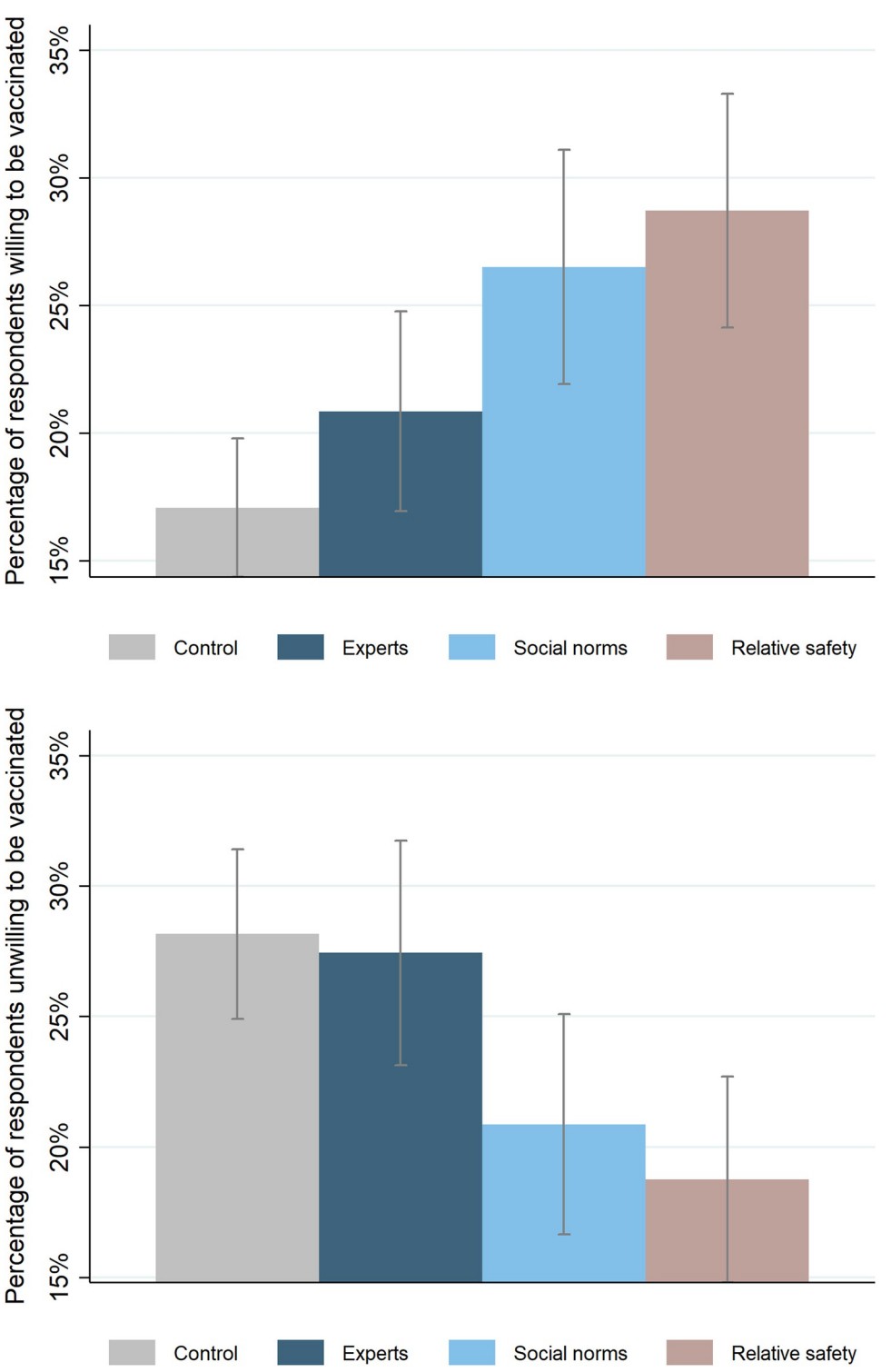

**Fig 5. Treatment effects on people's willingness and unwillingness to get vaccinated.** Notes: 95 percent confidence intervals are shown in error bars.

vaccination equilibrium, there is some hope in this finding. Vaccinating those who are already willing to be vaccinated may itself help to change the attitudes of others as they see their fellow community members safely taking the vaccine.

Third, health workers have a prominent role to play in increasing people's willingness to receive a vaccine for COVID-19. The phone survey illustrated that around half of all hesitant respondents said they would be more willing to be vaccinated if health workers recommended it. In addition, the failure of the expert advice treatment to shift people's views in the online experiment, suggests that generic advice from experts will not serve as an adequate substitute for specific matter-of-fact advice received from trusted local health workers. One potential complicating factor in PNG at least, is that vaccine hesitancy is comparatively high among health workers themselves [25]. This suggests that high priority should be given to educating local health workers about COVID-19 vaccines so they can become advocates in their community for vaccination. Also, there have been incidences of health workers being attacked when providing vaccinations in PNG [7–9]. This points to the need for a carefully staged 'educate, then vaccinate', approach, as well as the importance of security for vaccination teams.

Existing theoretical models of people's vaccination choices often include judgements of relative risk, in which people's actions are shaped be their feelings regarding the risks they believe both vaccine and disease pose [17]. The findings presented in this paper fit with a model of expectations based on judgements of relative risk. Specifically, the results of the phone survey and the online experiment show that low trust in the vaccine and concern about side effects appears to be a major driver of unwillingness to be vaccinated against COVID-19 in PNG. In the case of COVID-19 vaccines in PNG, our findings also demonstrate that expectations are not set in stone: information can change people's assessment of the relative risks and benefits associated with vaccinations and disease.

## Limitations and future work

There is good reason to believe the results from the phone survey will have external validity across PNG. Very careful sampling and weighting approaches were applied to ensure national coverage. There are also some grounds for confident that the experiment's findings will be externally valid too: a non-trivial share of Papua New Guinea's population uses Facebook, and resulting data were weighted carefully. Nevertheless, there remains the possibility that people participating in the experiment differ from the population of PNG as a whole in a manner that was both not captured by the sociodemographic traits that we used to weight our data, and which had a direct impact on treatment efficacy.

More importantly, both the survey and experiment asked people about their vaccination intentions: they did not measure whether people actually got vaccinated or not. For feasibility's sake, we were limited to focusing on surveyed intentions. Yet, work in the United States has shown stated vaccine intentions and actual practice can differ [26]. This is an acknowledged limitation.

Another limitation with the experimental findings is that, although they changed participants' views in the short-term, it is always possible that the effects of the information treatments waned over time.

Finally, in a country like PNG, online messages have a potentially useful role in changing people's views. Yet, the main task of increasing people's willingness to be vaccinated will fall to community health workers. There is always a chance that the efficacy of information is different when delivered in person by health workers, rather than online.

Together these limitations speak to an important future research program for PNG. More needs to be learnt, particularly about optimizing messages from health workers as the country

continues to roll out its community vaccination program. Experimental methods should be used to test which methods are most effective in changing attitudes, and which messages have the most durable positive impacts on beliefs, and actions, when delivered by the country's health workers.

For other developing countries, our findings provide a useful starting point for research into how attitudes to how COVID-19 vaccines can be changed. We have shown that there is considerable potential to use information to shift attitudes to COVID-19 vaccines in a country where vaccine hesitancy is very high. There is much scope to build on this insight with work conducted elsewhere.

## Supporting information

**S1 File. Survey weighting.**
(DOCX)

**S2 File. Survey questions about vaccine hesitancy used in analysis.**
(DOCX)

**S1 Fig. Preferred mode of vaccine information delivery (from phone survey).**
(DOCX)

**S2 Fig. Experiment treatment effects across demographic groups.**
(DOCX)

**S1 Table. Relationship between drivers of vaccine hesitancy and willingness to be vaccinated.**
(DOCX)

**S2 Table. Relationship between drivers of vaccine hesitancy and unwillingness to be vaccinated.**
(DOCX)

**S3 Table. Treatment effects with sociodemographic controls in regression models.**
(DOCX)

**S4 Table. Balance table across treatment groups in experiment.**
(DOCX)

## Author Contributions

**Conceptualization:** Christopher Hoy, Terence Wood, Ellen Moscoe.

**Data curation:** Christopher Hoy, Terence Wood, Ellen Moscoe.

**Formal analysis:** Christopher Hoy.

**Funding acquisition:** Christopher Hoy, Ellen Moscoe.

**Investigation:** Christopher Hoy, Terence Wood, Ellen Moscoe.

**Methodology:** Christopher Hoy, Terence Wood, Ellen Moscoe.

**Project administration:** Christopher Hoy, Ellen Moscoe.

**Resources:** Christopher Hoy, Ellen Moscoe.

**Software:** Christopher Hoy.

**Supervision:** Christopher Hoy, Ellen Moscoe.

**Validation:** Christopher Hoy.

**Visualization:** Christopher Hoy, Terence Wood.

**Writing – original draft:** Christopher Hoy, Terence Wood.

**Writing – review & editing:** Christopher Hoy, Terence Wood, Ellen Moscoe.

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
