## [Decision Letter · Decision Letter 0]

22 Aug 2022

PONE-D-22-11108

Addressing Vaccine Hesitancy in Developing Countries: Survey and Experimental Evidence

PLOS ONE

Dear Dr. Hoy,

Thank you for submitting your manuscript to PLOS ONE. After careful consideration, we feel that it has merit but does not fully meet PLOS ONE’s publication criteria as it currently stands. Therefore, we invite you to submit a revised version of the manuscript that addresses the points raised during the review process.

This paper is topical and both reviewers felt that it had merit.

It did, however, have a number of flaws. The reviewers felt that the introduction required more detail and the description of the actual study design was poor. In addition, there were frequent omissions when it came to abbreviations and statistical packages used etc.

We look forward to receiving your revised manuscript.

Kind regards,

Elizabeth S. Mayne, M.D.

Academic Editor

PLOS ONE

https://journals.plos.org/plosone/s/file?id=ba62/PLOSOne_formatting_sample_title_authors_affiliations.pdf".

“We are grateful for funding provided by Gavi, the Vaccine Alliance; the Korea Trust Fund for Economic and Peace-Building Transitions; and The United States Agency for International Development (through the Papua New Guinea UNICEF office).”

“Funding for this study was provided by Gavi, the Vaccine Alliance; the Korea Trust Fund for Economic and Peace-Building Transitions; and The United States Agency for International Development (through the Papua New Guinea UNICEF office).

The funders played no role in the the study design, data collection and analysis, decision to publish, or preparation of the manuscript.”

Additional Editor Comments:

This paper is topical and both reviewers felt that it had merit.

It did, however, have a number of flaws. The reviewers felt that the introduction required more detail and the description of the actual study design was poor. In addition, there were frequent omissions when it came to abbreviations and statistical packages used etc.

Reviewers' comments:

Reviewer's Responses to Questions

**Comments to the Author**

1. Is the manuscript technically sound, and do the data support the conclusions?

Reviewer #1: Yes

Reviewer #2: Partly

2. Has the statistical analysis been performed appropriately and rigorously? 

Reviewer #1: Yes

Reviewer #2: No

3. Have the authors made all data underlying the findings in their manuscript fully available?

Reviewer #1: Yes

Reviewer #2: Yes

4. Is the manuscript presented in an intelligible fashion and written in standard English?

Reviewer #1: Yes

Reviewer #2: Yes

5. Review Comments to the Author

Reviewer #1: This interesting manuscript on vaccine hesitancy in Papua New Guinea reports on collected data via two methods: (1) a phone survey of vaccine attitudes and their determinants, followed by (2) an online RCT conducted via Facebook, to measure the differential effects of one of four messages on participants' willingness and unwillingness to be vaccinated. Good care was taken to ensure that the survey and the experiment were representative of the population of Papua New Guinea. The authors are to be congratulated on an extremely well-written paper, devoid of any dense "academese" - it was a pleasure to read such clear writing.

There are a few issues that I feel may strengthen the paper if addressed:

MAJOR

The paper didn't really have a limitations section. Two important limitations are worth mentioning (in my opinion):

1. In the survey experiment, while I appreciate that quotas were applied based on demographic characteristics (age, gender and regions), since only 20% of the population use Facebook, the potential of selection bias is still significant. (E.g. perhaps Facebook users are more likely to be open to new experience, or more likely to be susceptible to social messaging, etc.)

2. There is a strong potential for the short-term effects seen in the experiment to wane with time. Participants appear to have had to reply almost immediately about their intentions, following different messages. However, we don’t know whether these changes in intentions are durable. Did participants feel the same the next day, or the week after? Or did the effects fade back to baseline? Given that this study can't answer this question (if I'm understanding it correctly), this is a significant potential limitation to the robustness of the findings in the real world.

MINOR:

1. The last three paragraphs of the introduction highlight the key results. I'd leave the results for the relevant section though, if possible.

Reviewer #2: Although the study is relevant, the authors did not research well about the subjects and methodology/tools used to study vaccine hesitancy before writing the manuscripts. The introduction and methodology is poorly written.

1. The introduction is scanty and does not give adequate background of the vaccine hesitancy. It must be re-written and include only introduction and not the method and results.

2. Page 3: Paragraph 3 and 4 should move to method and paragraph.

3. Page 4: paragraph 2-4 should move to results.

4. There are no line numbers used in the document and it make it difficult to review- the authors did not follow PLOS instructions guidelines.

5. Please provide additional details regarding participant consent in the text, if it is written or verbal, and if verbal, how it was documented and witnessed.

6. Survey Experiment-How did the author ensure that the author ensure that the participant included minors who used wrong age to join Facebook?

7. Statistical analysis: It is not clear which software/program was used for statistics. Please include a section on how the data was analysed and what was considered statistically significance.

8. The figures need to be correspond to the text eg. Figure 2 does not have any percentages while

9. It is not clear which study type was used: cross-sectional survey or convenience sampling etc

10. It is not clear how was the data collected, managed and captured and whether they used any Likert scale.

11. The authors need to be consistency if they are using PNG or Papua New Guinea.

12. All the abbreviations must be written in full before abbreviation.

13. The limitations of the study are not explicitly addressed in the manuscript.

14. All the abbreviations must be written in full before abbreviation.

6. PLOS authors have the option to publish the peer review history of their article (what does this mean?). If published, this will include your full peer review and any attached files.

Reviewer #1: No

Reviewer #2: No

---

## [Author Response · Author response to Decision Letter 0]

21 Sep 2022

We have provided a detailed response to reviewers in the document titled "2 Response to Reviewers".

---

## [Decision Letter · Decision Letter 1]

28 Oct 2022

Addressing Vaccine Hesitancy in Developing Countries: Survey and Experimental Evidence

PONE-D-22-11108R1

Dear Dr. Hoy,

We’re pleased to inform you that your manuscript has been judged scientifically suitable for publication and will be formally accepted for publication once it meets all outstanding technical requirements.

Kind regards,

Elizabeth S. Mayne, M.D.

Academic Editor

PLOS ONE

Additional Editor Comments (optional):

Reviewers' comments:

Reviewer's Responses to Questions

**Comments to the Author**

1. If the authors have adequately addressed your comments raised in a previous round of review and you feel that this manuscript is now acceptable for publication, you may indicate that here to bypass the “Comments to the Author” section, enter your conflict of interest statement in the “Confidential to Editor” section, and submit your "Accept" recommendation.

Reviewer #1: All comments have been addressed

Reviewer #2: All comments have been addressed

2. Is the manuscript technically sound, and do the data support the conclusions?

Reviewer #1: Yes

Reviewer #2: Yes

3. Has the statistical analysis been performed appropriately and rigorously? 

Reviewer #1: Yes

Reviewer #2: Yes

4. Have the authors made all data underlying the findings in their manuscript fully available?

Reviewer #1: Yes

Reviewer #2: Yes

5. Is the manuscript presented in an intelligible fashion and written in standard English?

Reviewer #1: Yes

Reviewer #2: Yes

6. Review Comments to the Author

Reviewer #1: (No Response)

Reviewer #2: The authors have addressed all the minor comments requested to my satisfaction. It is acceptable for publication

7. PLOS authors have the option to publish the peer review history of their article (what does this mean?). If published, this will include your full peer review and any attached files.

Reviewer #1: **Yes: **Jeremy Nel

Reviewer #2: No

---

## [Editor Report · Acceptance letter]

8 Nov 2022

PONE-D-22-11108R1 

Addressing Vaccine Hesitancy in Developing Countries: Survey and Experimental Evidence 

Dear Dr. Hoy:

I'm pleased to inform you that your manuscript has been deemed suitable for publication in PLOS ONE. Congratulations! Your manuscript is now with our production department. 

Kind regards, 

on behalf of

Dr. Elizabeth S. Mayne 

Academic Editor

PLOS ONE